# A Structured Recognition Method for Invoices Based on StrucTexT Model

**Zhijie Li \*, Wencan Tian, Changhua Li, Yunpeng Li and Haoqi Shi**

School of Information and Control Engineering, Xi'an University of Architecture and Technology, Xi'an 710055, China; twc858364@163.com (W.T.); lch304502@126.com (C.L.); liyunpeng696@163.com (Y.L.); danhoo598@163.com (H.S.)
\* Correspondence: lizhijie@xauat.edu.cn; Tel.: +86-15529037880

**Abstract:** Invoice recognition has long been an active research direction in the field of image recognition. Existing invoice recognition methods suffer from a low recognition rate for structured invoices, a slow recognition speed, and difficulty in mobile deployment. To address these issues, we propose an invoice-structured recognition method based on the StrucTexT model. This method uses the idea of knowledge distillation to speed up the recognition speed and compress the model size without reducing the model recognition rate; this is achieved using the teacher model StrucTexT to guide the student model StrucTexT_slim. The method can effectively solve the problems of slow model recognition speed and large model size that make mobile deployment difficult with traditional methods. Experimental results show that the proposed model achieves an accuracy rate of over 94% on the SROIE and FUNSD public datasets and over 95% on the self-built structured invoice dataset. In addition, the method is 30% faster than other models (YOLOv4, LeNet-5, and Tesseract-OCR) in terms of recognition speed, while the model size is compressed by about 20%.

**Keywords:** structured recognition of invoices; invoice layout analysis; pre-training; knowledge distillation

## 1. Introduction

Optical Character Recognition (OCR) is currently one of the most widely used visual artificial intelligence technologies. With the rapid development of multimedia information based on images, image text recognition technology has become an important medium for information transmission. This technology can automatically locate, segment, and recognize the structured text content contained in images and has important value for the automatic understanding of image semantics, image retrieval, and indexing [1]. In the field of office automation, the number of expense reimbursement documents processed by large enterprises is increasing rapidly due to the rapid development of social, economic, and business needs. Therefore, the optimal way in which we can apply advanced information technology to handle the large number of invoices generated during daily production has become a hot topic in the field of text recognition.

OCR technology, driven by strong market demand, has been applied in the financial sector and has recently become a hot topic in the field of text recognition research. Research on invoice recognition has been previously conducted in foreign countries, especially in OCR technology research institutions in the United States, Canada, Japan, and other countries. As research progresses, foreign scholars have also developed many practical products. For example, MiTek Systems' CheckQuest product has been used by banks such as the Bank of Thayer and Mount Prospect National Bank, and A2iA's CheckReader has been applied to commercial banks in the United States and France.

Structured recognition of invoices has been carried out in China. Tang J. et al. [2] proposed a method for the structured recognition of value-added tax invoice information using HRNet and YOLOv4. The method showed strong recognition ability with value-added tax invoices but had limitations in recognizing other types of invoices such as taxi

receipts and flight itineraries. Yin Z. J. et al. [3] proposed an improved invoice recognition method based on modifying the activation function and increasing the number of feature maps in the LeNet-5 convolutional neural network to improve network performance; however, its structured output effect is not ideal. Sun R. B. et al. [4] proposed a complex invoice adaptive recognition method based on Tesseract-OCR, which achieved adaptive adaptation of table headers and content through the extraction of table positions and customized templates within a structured output. However, its adaptability to large quantities of complex invoices is poor, as each new type of invoice requires analysis from scratch.

In the common process of invoice reimbursement, the region of interest (ROI) that needs to be identified on an invoice is commonly divided into four parts: purchaser information, seller information, amount, and basic invoice information.

Figure 1 shows a value-added tax invoice in China. In the reimbursement process, the ROIs that need to be identified are the purchaser's name, the taxpayer's identification number, the invoice code, invoice number, billing date, amount, etc. Here is an example code that illustrates how traditional invoice recognition methods store the recognized segments as strings and then use regular expression-matching functions to obtain the corresponding ROI segments:

re.findall ('\d* 年 \d* 月 \d* 日',txt)

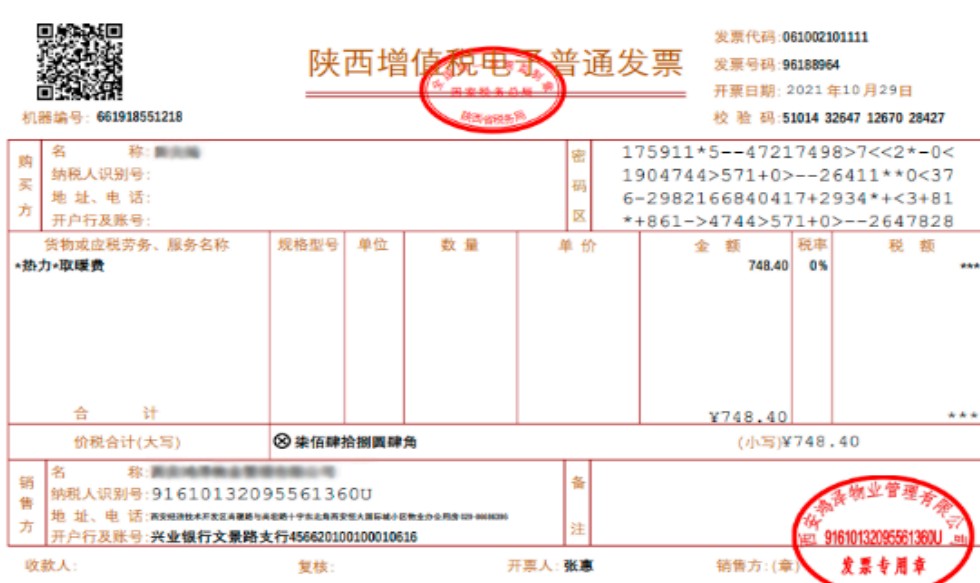

**Figure 1.** A VAT invoice.

In the above code, the character "年" represents the year, the character "月" represents the month, and the character "日" represents the day. The above code uses the regular expression function to return the "billing date" ROI. However, there are some issues with this approach. It is only suitable for segments such as the billing date and invoice code, which have a fixed style and a relatively simple format. Its recognition ability is poor for more complex segments, such as the purchaser's address, which lack obvious formats. In addition, it may have difficulty distinguishing between the purchaser's and seller's information, thereby leading to confusion.

The aforementioned invoice recognition methods may be competent for general invoice recognition tasks but may also reveal some problems. The first lies in the recognition speed of the model. In daily production, the number of invoices that need to be recognized is often huge, and the recognition speed of the model will affect its working efficiency.

Secondly, high-precision models usually lead to a very large model size, which places great demands on the running equipment and is not easy to deploy on a mobile device.

To address the limitations of the aforementioned methods, the article proposes a structured invoice recognition method based on the StrucTexT model. The method builds on open-source OCR algorithms and can output recognized information in a structured format, solving the problem of mismatched key segments and recognition information. It not only focuses on value-added tax invoices but also has high recognition accuracy for other reimbursement documents such as taxi receipts and flight itineraries, solving the problem of a limited range of recognized documents and poor usability. The model is also optimized for size and can be deployed on edge devices, making it easier to operate on mobile devices. The method compresses the model's size, speeds up its recognition, and improves its efficiency. This method achieves automatic recognition, entry, and storage of invoice information, saving time for financial personnel and streamlining the financial reimbursement process.

This paper is divided into four main chapters. The first section is an introduction, which mainly analyzes the current status of invoice recognition-related research, summarizes the shortcomings of existing methods, and proposes a structured recognition method for invoices based on StrucTexT. The second section mainly introduces the base model used, StrucTexT, and optimizes the model by knowledge distillation; the third chapter is an experimental analysis in which the optimized model is compared with other models on a publicly available dataset. The third section is an experimental analysis, in which the optimized model is compared with other latest models on the public dataset, and the identification experiments of key segments of bills are conducted on the self-built dataset to illustrate the effectiveness and novelty of the method. Section 4 is the conclusion, which mainly gives a summary of the work in this paper.

## 2. Related Models and Optimization

### 2.1. StrucTexT

OCR structuring technology often addresses two high-frequency application task types: entity labeling and entity linking. Entity labeling refers to the extraction of text content from OCR recognition results corresponding to predefined entity labels, such as "Amount", "Purchaser Name", etc. Entity linking refers to analyzing the relationship between text entities, for example, whether they can form key_value pairs or whether they belong to the same row or column in a table.

In invoice recognition, invoices are visual, text-rich images with multiple attributes such as text, image, and layout. Multimodal cues are used for modeling, and Chinese and English segment-level multimodal features are incorporated into structured OCR pre-training for feature enhancement.

Figure 2 shows a schematic diagram of the overall model framework of the model StrucTexT [5] used in this paper. In a given input image with pre-processed OCR results such as bounding boxes and text segment content, for example, various pieces of information in terms of text, image, and layout are utilized through the feature-embedding phase. The multimodal embedding is then fed back to the pre-trained transformer to obtain rich semantic features. The transformer accomplishes cross-modal fusion by creating interactions between different modal inputs. Figure 3 shows a diagram of the model's pre-training task. Finally, the structured text understanding module receives the encoded features, performs entity recognition for entity labeling, and then performs relationship extraction of entity connections, as shown in Figure 4.

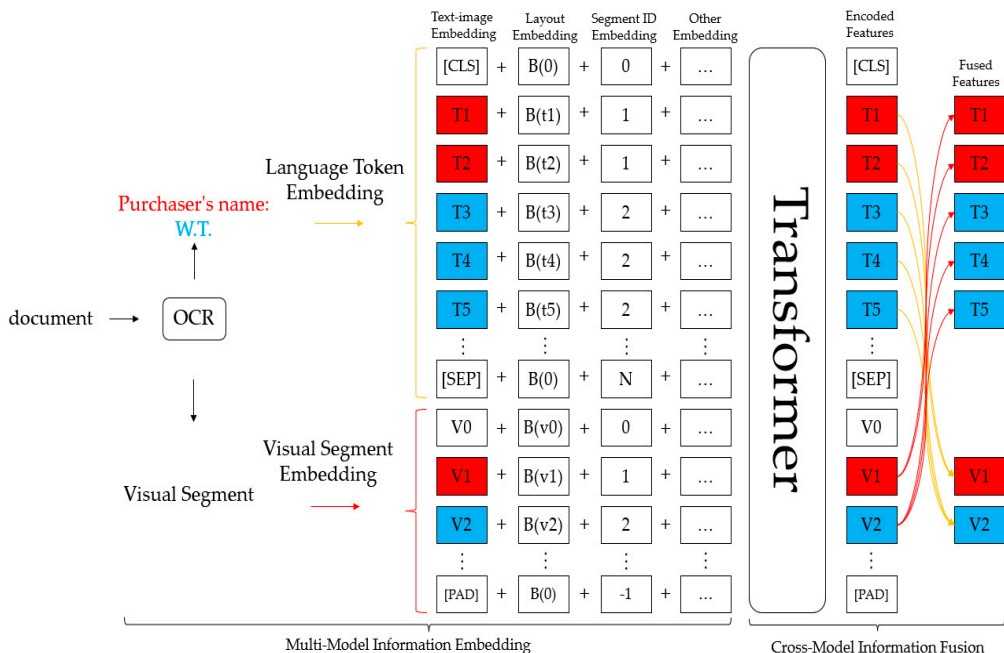

**Figure 2.** An overall illustration of the model framework of StrucTexT.

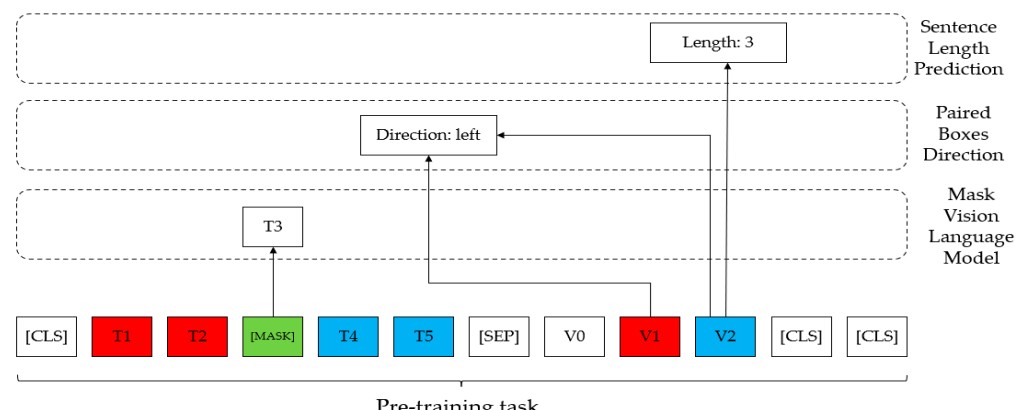

**Figure 3.** Pre-training tasks.

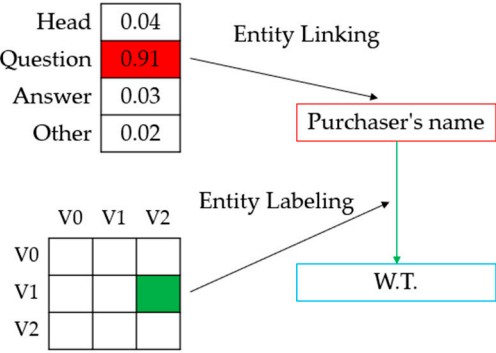

**Figure 4.** Structured text understanding.

Given a document image $I$ with $n$ text segments, StrucTexT carries out open-source OCR algorithms to obtain the $i$-th segment region with the top_left and bottom_right bounding boxes $b_i = (x_0, y_0, x_1, y_1)$, and its corresponding text sentence $t_i = \left\{ c_1^i, c_2^i, \ldots, c_{l_i}^i \right\}$,

where $c$ is a word or character and $l_i$ is the length of $t_i$. For every segment or word, StrucTexT uses the encoded bounding boxes as their layout information.

$$L = Emb_l(x_0, y_0, x_1, y_1, w, h) \tag{1}$$

where $Emb_l$ is a layout embedding layer and $w$ and $h$ are the shapes of the bounding box $b$. It is worth mentioning that we estimate the bounding box of a word using its own text segment, considering OCR results that lack word-level information.

Following a common method, StrucTexT utilizes WordPiece [6] to tokenize text sentences. After that, all of the text sentences are gathered as a sequence $S$ by sorting the text segments from the top_left to the bottom_right. Intuitively, a pair of special tags $[CLS]$ and $[SEP]$ are added at the beginning and end of the sequence, as $t_0 = \{[CLS]\}$ and $t_{n+1} = \{[SEP]\}$. Thus, StrucTexT can define the language sequence $S$ as follows:

$$\begin{aligned} S &= \{t_0, t_1, \ldots, t_n, t_{n+1}\} \\ &= \left\{[CLS], c_1^1, \ldots, c_{l_1}^1, \ldots, c_1^n, \ldots, c_{l_n}^n, [SEP]\right\} \end{aligned} \tag{2}$$

Then, StrucTexT sums the embedded feature of $S$ and the layout embedding $L$ to obtain the language embedding $T$:

$$T = Emb_t(S) + L \tag{3}$$

where $Emb_t$ is a text-embedding layer.

Within the model's architecture, StrucTexT uses ResNet50 [7] with FPN [8] as the image feature extractor to generate feature maps of $I$. Afterwards, the image feature of each text segment is extracted from the $CNN$ maps using RoIAlign [9], according to $b$. The visual segment embedding $V$ is computed as

$$V = Emb_v(ROIAlign(CNN(I), b)) + L \tag{4}$$

where $Emb_v$ is the visual embedding layer. Furthermore, the entire feature map of image $I$ is embedded as $V_0$, in order to introduce global information into image features.

Compared with the vision_language tasks based on wild pictures, understanding the structured document requires higher semantics to identify the ambiguous entities. Thus, StrucTexT proposes segment ID embedding $S^{id}$ to allocate a unique number to each text segment, its image, and its text features, which will then create an explicit alignment of cross-modality clues.

In addition, StrucTexT adds two other embeddings to the input. The position embedding $P^{id}$ encodes the indexes from 1 to the maximum sequence length, and the segment embedding $M^{id}$ denotes the modality for each feature. All of the above embeddings have the same dimensions. In the end, the input of the StrucTexT model is represented as the combination of the embeddings.

$$Input = Concat(T, V) + S^{id} + P^{id} + M^{id} \tag{5}$$

Moreover, StrucTexT appends several [PAD] tags to make the short input sequence a fixed length. An empty bounding box with zeros is assigned to the special [CLS], [SEP], and [PAD] tags.

StrucTexT collects multi-modal information from visual segments, text sentences, and position layouts to produce an embedding sequence. We support image_text alignment between different granularities by leveraging the segment IDs mentioned above. At this stage, we introduce a transformer network to encode the embedding sequence and establish deep fusion between modalities and granularities. Crucially, during the pre-training stage, three self-supervised tasks encode the input features to learn task-agnostic joint representations. The details of these tasks are introduced below, and the patterns of all

self-supervised tasks are shown in Figure 5. The Chinese characters in the Figure 5 are visual segments in the invoice.

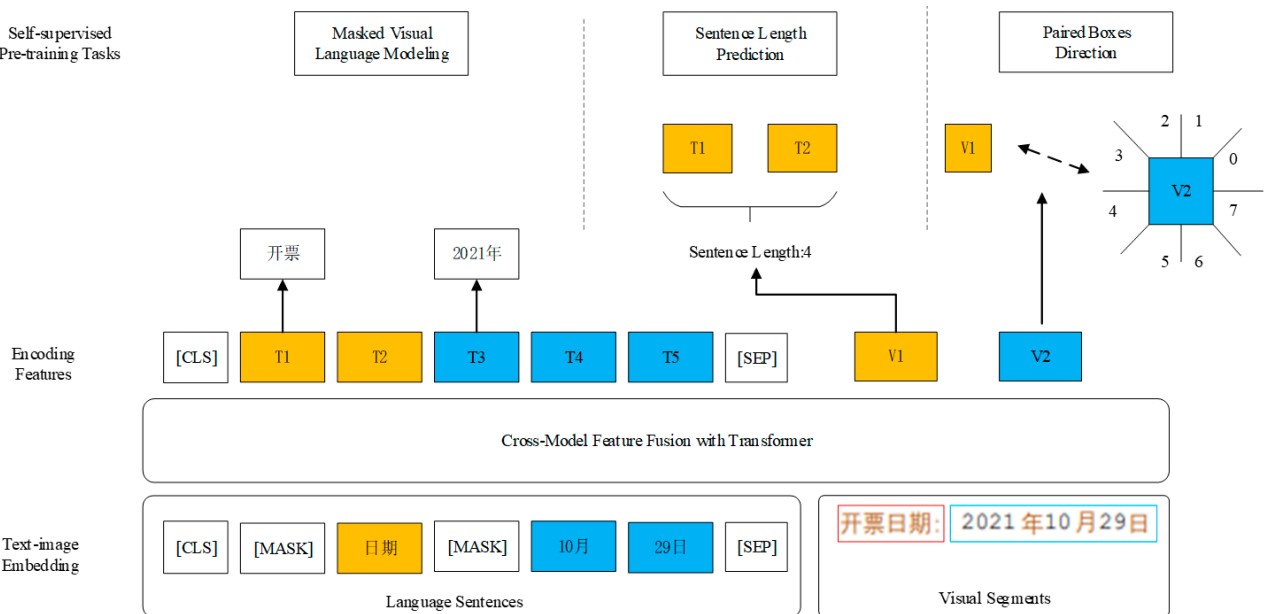

**Figure 5.** Self-supervised tasks.

In the following section, the three self-supervised tasks of masked visual language modeling, sentence length prediction, and paired box direction are described separately.

(a) Masked visual language modeling (MVLM)

This task is used to facilitate the capture of contextual representations in terms of language. Following the masked multimodal modeling model of ViLBERT, the model selects about 15% of the tokens from the language sequence, masks 80% of them with [MASK] tags, replaces 10% of them with random tokens, and keeps 10% of the tokens unchanged. The model is required to reconstruct the corresponding notation. The model does not follow the masking of ViLBERT's image region precisely, but it retains all information and encourages the model to look for cross-modal cues whenever possible.

(b) Sentence length prediction (SLP)

This task is used to mine fine-grained semantic information on the image side. SLP requires the model to recognize the length of the segment from each visual feature. In this way, the encoder is forced to learn image features using the same segment ID, and more importantly, it is also forced to acquire linguistic sequence knowledge. This information flow accelerates the deep cross-modal fusion of textual, visual, and layout information.

In addition, to avoid the interference caused by the generation of sub-words, the model only counts the first occurrence of sub-words in order to keep the length between the language sequence and the image fragment the same. Thus, the model simply and efficiently creates additional alignment between the two granularities.

(c) Paired box direction (PBD)

As a third self-supervised task, PBD aims to exploit global layout information. The purpose of the PBD task is to learn the integrated geometric topology of the document structure by predicting the pairwise spatial relationships between text segments. First, a 360° region is divided into eight identical buckets. Second, the angle $\theta_{ij}$ between text segments $i$ and $j$ is computed and labeled with one of the buckets. Next, subtraction is performed between two visual features on the image side in order to obtain the result $\Delta \hat{V}_{ij}$ as input to the PBD:

$$\Delta \hat{V}_{ij} = \hat{V}_i - \hat{V}_j \tag{6}$$

where the symbol "^" is used to indicate the features after transformer encoding. $\hat{V}_i$ and $\hat{V}_j$ express the visual features of the *i*-th segment and *j*-th segment.

Finally, PBD is defined as a classification task that uses $\Delta\hat{V}_{ij}$ to determine the relative position and orientation.

### *2.2. Optimization*

According to the pre-training task of the "masked visual language model" in the invoice-related document images and the StrucTexT model in the domestic scene, the pre-training task is extended to the segment level. By randomly selecting some segments and masking all tokens in the segment, the model is required to restore these tokens in order to learn more complex semantic information. The original model uses the masked visual language model, which can only learn relatively simple semantic information. For rich visual text images such as value-added tax special invoices, the semantic information is relatively complex. By extending the pre-training task using the masked segment prediction, the model's understanding of complex semantic information is improved, as is its ability to understand and recognize various pieces of complex invoice information.

Meanwhile, it should be noted in the use of the model that although the StrucTexT model has excellent performance and a fast recognition speed for single-ticket recognition, in practical application scenarios, said tickets are often numerous and disorderly. StrucTexT contains 107 M parameters, and although the model's performance is high, the complexity of the model is also high, and the cost of applying large models in practical scenarios is high. Considering that the final recognition result still needs to be verified by financial personnel, the accuracy and speed of the model must be balanced; the recognition speed should be improved without sacrificing the recognition accuracy.

To compress the model and reduce its parameter count, there are several main methods currently available, such as pruning [10], quantization [11], knowledge distillation [12,13], and so on. Among them, knowledge distillation uses a teacher model to guide the student model to learn specific tasks. As a result, the student model has performance comparable to that of the teacher model but with a significantly reduced parameter count, thereby achieving model compression and acceleration. The method uses the StrucTexT model as the teacher model to guide the student model, StrucTexT_slim, which is essentially mutual supervision between the output and feature maps. The teacher model is needed to load the pre-trained model and fix the parameters.

To obtain a softer classification output, the knowledge distillation algorithm adds a temperature coefficient *T* to the softmax function in order to soften the category probability of the target, which is calculated as shown:

$$q_i = \frac{exp(z_i/T)}{\sum\limits_{j} exp(z_j/T)} \tag{7}$$

where $z_i$ is the content of the output layer. The temperature coefficient *T* is taken as 1, which is the original softmax function. As *T* decreases, the relative size of the probability of incorrect categories becomes more apparent; as T increases, the probability distribution of the output target category becomes smoother, and the softness of the label increases. Figure 6 shows the algorithm flowchart for knowledge distillation.

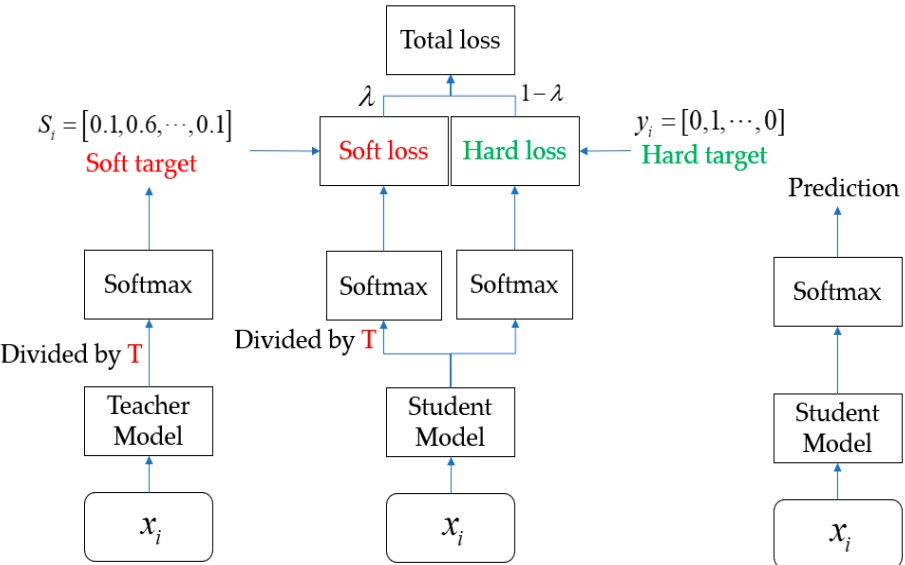

**Figure 6.** Knowledge distillation algorithm flowchart.

For the same input vector $x_i$, the predicted output of the teacher model is calculated, divided by the temperature coefficient T, and then transformed via softmax to obtain the soft target, i.e., the softened probability distribution. Afterwards, the predicted output of the student model is calculated, divided by the temperature coefficient T, and transformed via softmax, and cross-entropy loss is calculated using the soft target. The overall loss calculation is as follows:

$$Loss_{total} = \lambda Loss_{soft}(p_s, p_t) + (1 - \lambda)Loss_{hard}(p_s, y) \tag{8}$$

The overall loss, $Loss_{total}$, is divided into two parts, $Loss_{soft}$ and $Loss_{hard}$, which are calculated as follows:

$$Loss_{soft}(p_s, p_t) = -\sum p_s log p_t \tag{9}$$

$$Loss_{hard}(p_s, y) = -\sum p_s log y \tag{10}$$

where $p_s$ and $p_t$ represent the predicted output of the student model and the teacher model, respectively; $y$ is the true label and $\lambda$ is the weighting factor. During the early stages of training, the value of $\lambda$ can be increased in order to quickly enhance the student model's ability to identify simple samples, ensuring that the student network can quickly acquire the 'knowledge' of the teacher model. In the later stages of training, the $\lambda$ value can be appropriately reduced to improve the student model's ability to identify difficult samples while combining the true labels (i.e., $y$) to help the student model with recognition.

The final distillation training loss function contains the following five components:

(1) The CTC branch of the final output (head_out) of student and teacher, with a CTC loss of gt, weighted by 1. Here, because both sub-networks need to update their parameters, both have to calculate the loss using g.

(2) The SAR branch of the student's and teacher's final output (head_out), with a SAR loss of gt, weighted by 1. Here, both need to calculate the loss using g, because both sub-networks need to update their parameters.

(3) The DML loss between the CTC branches of the student's and teacher's final output (head_out), with a weight of 1.

(4) The DML loss between the student and the SAR branch of the teacher's final output (head_out), with a weight of 0.5.

(5) The l2 loss between the student's and teacher's backbone network output (backbone_out) has a weight of 1.

In terms of model structure, in order to add more loss functions and to ensure the scalability of the distillation method, the output of each sub-network is saved as a dict, which contains the sub-module outputs. In the model, the output of each sub-network is a dict; the key contains backbone_out, neck_out, and head_out; and the value is the tensor of the corresponding module.

For post-processing, the output of these two sub-networks is extracted and decoded for the prediction results of the distillation model. The CTC-decoded outputs of both the student's and teacher's sub-networks are computed, and a dict is returned, with the key being the name of the sub-network used for processing and the value being the list of sub-networks used for processing.

RecMetric is set as the base class for metric calculation, and eventually, the "acc" metrics of the Student's sub-network are used as the judgment metric for saving the best model.

## 3. Experiment and Analysis

### 3.1. Dataset

The datasets used for the experiments include the public datasets SROIE and FUNSD and self-built datasets.

SROIE is a public dataset for ticket information extraction that is provided by the ICDAR 2019 Challenge. It contains 626 training ticket data and 347 test ticket data, each containing the following four predefined segments: company name, date, address, and total price.

FUNSD is a dataset for form understanding, containing 199 fully labeled scanned images of marketing reports, advertisements, and academic reports; it is divided into 149 training sets and 50 test sets.

The self-built dataset consists of 100 VAT invoices and airline itineraries obtained from the university's finance office and used by individuals; it is fully labeled using open-source dataset-labeling software.

### 3.2. Experimental Environment and Parameter Settings

The experimental environment is based on Win10, using an Intel i7-11800F CPU, an NVIDIA Geforce RTX 3090 24 G GPU, 16 GB of RAM, Python 3.6 + CUDA 10.2, and the Pycharm community IDE.

The models compared in the experiments are LayoutLM [14] and LayoutLMv2 [15], which are pre-trained document-understanding models that act by jointly modeling the layout and text and image multimodel information of a document using the spatial relative attention mechanism, newly introducing text and image that are related. Among them, the BASE model is a 12-layer transformer; each layer contains 768 implied units and 12 attention heads, with 113 M parameters. The LARGE model is a 24-layer transformer; each layer contains 1024 implied units and 16 attention heads, with 343 M parameters.

The experiments follow a typical pre-training and fine-tuning strategy by rescaling and padding the images to a size of $512 \times 512$, with the input sequence set to a maximum length of 512. A 12-layer transformer with 768 implicit units and 12 attention heads is selected (the same as the BASE model of LayoutLM). The batch size is set to 4, the learning rate is [0.0001, 0.00001], and the knowledge distillation temperature coefficient T is set to 3.

### 3.3. Comparison of Experimental Results

The experiments were first conducted on the SROIE public dataset, and the optimized model StrucTexT_slim's performance in entity labeling was compared with that of LayoutLM, LayoutLMv2, YOLOv4, LeNet-5, and Tesseract-OCR. The results are shown in Table 1.

**Table 1.** Model performance comparison on SROIE.

| Model | Precision | Recall | F1 | Parameters | Time |
|---|---|---|---|---|---|
| LayoutLM_BASE | 0.944 | 0.944 | 0.944 | 113 M | 4.5 s |
| LayoutLM_LARGE | 0.952 | 0.952 | 0.952 | 343 M | 2.7 s |
| LayoutLMv2_BASE | 0.963 | 0.963 | 0.963 | 200 M | 3.6 s |
| LayoutLMv2_LARGE | 0.966 | 0.966 | 0.966 | 426 M | 2.1 s |
| StrucTexT | 0.967 | 0.968 | 0.969 | 107 M | 6.7 s |
| YOLOv4 [2] | 0.954 | 0.954 | 0.954 | 145 M | 4.6 s |
| LeNet-5 [3] | 0.941 | 0.942 | 0.941 | 92 M | 5.6 s |
| Tesseract-OCR [4] | 0.957 | 0.958 | 0.957 | 101 M | 7.3 s |
| StrucTexT_slim | 0.947 | 0.948 | 0.947 | 80 M | 4.1 s |

The time in the table indicates the average time required to recognize an image. It is obvious from the table that the improved model StrucTexT_slim has an accuracy of 94.7% on the SROIE dataset, which is 2% lower than that of the original model but second only to LayoutLMv2_Large in terms of recognition time. It also has the smallest model size. The comparison results on the FUNSD dataset are shown in Table 2.

**Table 2.** Model performance comparison on FUNSD.

| Model | Precision | Recall | F1 | Parameters | Time |
|---|---|---|---|---|---|
| LayoutLM_BASE | 0.760 | 0.816 | 0.787 | 113 M | 4.1 s |
| LayoutLM_LARGE | 0.760 | 0.822 | 0.790 | 343 M | 2.9 s |
| LayoutLMv2_BASE | 0.803 | 0.854 | 0.828 | 200 M | 3.7 s |
| LayoutLMv2_LARGE | 0.832 | 0.852 | 0.842 | 426 M | 2.0 s |
| StrucTexT | 0.857 | 0.810 | 0.831 | 107 M | 6.5 s |
| YOLOv4 [2] | 0.837 | 0.852 | 0.841 | 145 M | 4.8 s |
| LeNet-5 [3] | 0.821 | 0.819 | 0.825 | 92 M | 6.1 s |
| Tesseract-OCR [4] | 0.836 | 0.841 | 0.837 | 101 M | 7.5 s |
| StrucTexT_slim | 0.835 | 0.796 | 0.816 | 80 M | 3.9 s |

In the FUNSD dataset, entities belonging to other categories were ignored, and the average performance of the three categories (title, question, and answer) was used as the final result. Although the StrucTexT_silm model is not optimal in all metrics, the best-performing LayoutLMv2_LARGE model is only 2.6% higher in F1 score than the StrucTexT_silm model. Meanwhile, the LayoutLMv2_LARGE model is larger, consisting of a 24-layer and 16-head transformer that contains 426 M parameters. In addition, the LayoutLMv2_LARGE model uses 11M documents for pre-training. During the product's actual application and deployment, StrucTexT_silm performs significantly better.

The experiments also used the self-built dataset mentioned in Section 3.1 to recognize some keywords in VAT invoices and airline trip tickets in order to test the model's matching of keyword segments with the recognized information. The recognition results are shown in Table 3, and the model performs better in recognizing tickets such as VAT invoices and airline trip tickets.

As can be seen from the table, the recognition rate of the model for each key area of the VAT invoices is over 95%, which meets the requirements of invoice recognition.

From the above experiments, it can be concluded that the optimized model StrucTexT_slim has a higher accuracy; its recognition speed is greatly accelerated, and its model volume is compressed, which solves a series of problems (slow recognition speed, large model volume, and difficulty of deployment within a traditional invoice recognition model). The experiments also show that the model has high recognition accuracy in its assessment of complex invoices, such as VAT invoices.

**Table 3.** Keyword recognition performance.

| Keyword | Precision |
|---|---|
| Pre-tax amount | 1 |
| Tax rate | 1 |
| Tax amount | 1 |
| Name (seller) | 0.98 |
| Taxpayer identification number (seller) | 0.99 |
| Address telephone (seller) | 0.94 |
| Bank and account number (seller) | 0.93 |
| Name (purchaser) | 0.98 |
| Taxpayer identification number (purchaser) | 0.99 |
| Address telephone (purchaser) | 0.94 |
| Bank and account number (purchaser) | 0.93 |
| Invoice code | 0.99 |
| Invoice number | 0.99 |
| Date | 1 |
| Passenger name | 0.95 |
| Departure place | 0.99 |
| Destination | 0.99 |
| Flight number | 0.98 |
| E-ticket number | 0.97 |

## 4. Conclusions

In this article, a structured invoice recognition method based on the StrucTexT model is proposed. To address the issue of invoices with multiple attributes, the structured StrucTexT model is used for recognition but is made lightweight and therefore optimized for practical applications. The experimental results show that the proposed method achieves an accuracy rate of over 94% on the publicly available datasets SROIE and FUNSD and an accuracy rate of over 95% on a self-built structured invoice dataset, with a 30% decrease in average processing time. The model's size was reduced by 20%, making it more deployment-friendly. The proposed method has a fast recognition speed and is highly practical, all without sacrificing accuracy, making it applicable to production on a daily basis.

**Author Contributions:** Conceptualization, Z.L. and W.T.; methodology, Z.L. and W.T.; experiment design, Z.L. and W.T.; experimental validation, Z.L., W.T., Y.L. and H.S.; formal analysis, Z.L., W.T. and C.L.; resources, Z.L.; data collection, Z.L. and C.L.; writing—original draft preparation, W.T.; writing—review and editing, Z.L.; supervision, Z.L. and C.L.; project administration, Z.L. and C.L.; funding acquisition, Z.L. and C.L. All authors have read and agreed to the published version of the manuscript.

**Funding:** This research was funded by the National Key Research and Development Program of the 13th Five-Year Plan (2019YFD1100904), the National Natural Science Foundation of China (51878536), and the Science and Technology Program for Housing and Urban-Rural Development of Shaanxi Province (2020-K09).

**Institutional Review Board Statement:** Not applicable.

**Informed Consent Statement:** Not applicable.

**Data Availability Statement:** The public dataset SROIE is available at the following link: GitHub—zzzDavid/ICDAR-2019-SROIE: ICDAR 2019 Robust Reading Challenge on Scanned Receipts, OCR, and Information Extraction. The public dataset FUNSD is available at the following link: FUNSD (guillaumejaume.github.io) (accessed on 4 July 2022). Self-built datasets cannot be made public due to the privacy of invoices.

**Acknowledgments:** We are grateful for the financial support provided and the valuable comments from the editors and reviewers.

**Conflicts of Interest:** The authors declare no conflict of interest.

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
