# Peer review of "A Structured Recognition Method for Invoices Based on StrucTexT Model"

_applsci, doi:10.3390/app13126946_

Round 1
Reviewer 1 Report
1. In the abstract part, the novelty and key idea of the proposed method should be described. please address the novelty and key idea are not clear.
2. The problem definition of this work is not clear. In Sect. 1, the drawbacks of each conventional technique should be described one by one. The authors should emphasize the difference with other methods to clarify the position of this work further.
3. The effectiveness of this work is not clear. Through experiments, the authors must justify the effectiveness of the proposed method by comparing with the state-of-the-art methods. However, detailed comparison is shown with these techniques. The authors should show comparison data.
4. what the outcome of the proposed approach if the quality of invoice image is very low
Moderate editing of English language is required
Reviewer 2 Report
The manuscript is well-organized.
Also, some suggestions are provided, in case the authors consider them interesting to carry out.
Abstract and conclusion needs to improve.
Figure resolution needs to improve.
References need to increase.
The objective of the paper must be clear in the introduction.
The organization of paper section needs to add end the introduction.
Reviewer 3 Report
The article proposes a structured invoice recognition method based on the StrucTextT model. The method can be deployed on edge devices and used on mobile devices. StrucTextT is a flexible platform that uses visual language modeling and sentence length prediction and paired box direction tasks to incorporate multimodal information into text, image, and layout. However, although the performance of the StrucTexT model is high, its complexity is also high and the authors propose to make some improvements. A hidden segment prediction is added to improve the semantic understanding of the model at the segment level. In addition, knowledge distillation is used to lighten the model and improve invoice recognition speed.
The text is clear and the content is very understandable, The gain obtained after simplification of the model is impressive. However two questions arise:
· The FUNSD database seems small: very few learning and especially test samples. Is this sufficient to cover the maximum number of cases knowing that there are many fields to recognize
· Have you ported this system to a mobile and does the system keep the same performance?
Round 2
Reviewer 1 Report
Accept in present form
English language fine. No issues detected